# Gathering Big Data in Wireless Sensor Networks by Drone ^†^

**DOI:** 10.3390/s20236954

**Published:** 2020-12-05

**Authors:** Josiane da Costa Vieira Rezende, Rone Ilídio da Silva, Marcone Jamilson Freitas Souza

**Affiliations:** 1Departamento de Computação, Universidade Federal de Ouro Preto, Rua Diogo de Vasconcelos, 122, Bairro Pilar, Ouro Preto 35400-000, Brazil; josianecvieira@gmail.com (J.d.C.V.R.); marcone@ufop.edu.br (M.J.F.S.); 2Departamento de Tecnologia em Engenharia Civil, Computação, Automação, Telemática e Humanidades, Universidade Federal de São João Del Rei, Campus Alto Paraopeba—C.A.P, Rod.: MG 443, KM 7, Ouro Branco 36420-000, Brazil

**Keywords:** wireless sensor network, WSN, mobile sink, path planning, UAV

## Abstract

The benefits of using mobile sinks or data mules for data collection in Wireless Sensor Network (WSN) have been studied in several works. However, most of them consider only the WSN limitations and sensor nodes having no more than one data packet to transmit. This paper considers each sensor node having a relatively larger volume of data stored in its memory. That is, they have several data packets to send to sink. We also consider a drone with hovering capability, such as a quad-copter, as a mobile sink to gather this data. Hence, the mobile collector eventually has to hover to guarantee that all data will be received. Drones, however, have a limited power supply that restricts their flying time. Hence, the drone’s energy cost must also be considered to increase the amount of collected data from the WSN. This work investigates the problem of determining the best drone tour for big data gathering in a WSN. We focus on minimizing the overall drone flight time needed to collect all data from the WSN. We propose an algorithm to create a subset of sensor nodes to send data to the drone during its movement and, consequently, reduce its hovering time. The proposed algorithm guarantees that the drone will stay a minimum time inside every sensor node’s radio range. Our experimental results showed that the proposed algorithm surpasses, by up to 30%, the state-of-the-art heuristics’ performance in finding drone tours in this type of scenario.

## 1. Introduction

Wireless Sensor Network (WSN) is a computer network composed of small devices called sensor nodes. These devices are spatially distributed to monitor physical or environmental conditions to detect some phenomena located around them. They communicate by radio and can relay packets received from other nodes to forward them to a special node called sink, which is the interface with the users. Hence, these nodes create multihop networks to support many applications that require unattended operations. Traditionally in the literature, the major challenge for this kind of network is collecting, gathering, storage, and processing sensed data in an energy-efficient way since each sensor node has a battery as its energy supply [1]. A well-known strategy to reduce energy consumption and extend the WSN lifetime is adopting a mobile collector. As defined here, the term mobile collector is correlated to the terms mobile sink [2] or data mule [3] found in the literature. It consists of a particular node able to move within the area monitored by the WSN to gather data from the sensor nodes. Compared with the fixed collector, mobile collectors better distribute the sensor nodes’ energy consumption, avoiding premature disconnections. Furthermore, it decreases the energy consumption since the average size of the path followed by each data packet is reduced. The literature presents several works that consider mobile collectors in WSNs, such as [4,5,6,7,8,9,10,11].

A WSN can generate a very large volume of data due to the high number of devices scattered across vast geographic areas or the relatively significant volume of data stored in each sensor node [8]. According to [9], this is called Big Data, and the traditional data processing algorithms are inadequate to manipulate it. Despite our research efforts, almost all studies in the literature do not guarantee that the mobile collector will stay a minimum time covered by every sensor node’s radio to receive a large volume of data. Furthermore, the analyzed works only consider WSN limitations. The mobile collectors have no restrictions; that is, paths followed by these collectors have no length limitation. The collector can stop at any place inside the monitored area, and they have no time limitation to complete their trips for data gathering.

This work considers a drone with hovering capability (such as quad-copters) as a mobile collector and WSNs composed of sensor nodes with many data packets to transmit to the mobile collector (Big Data), as in [12]. The problem studied here is how to find a drone tour that minimizes the data gathering time. We focus on finding drone trajectories to reduce the drone’s time to gather all data stored in the sensor node memories. This type of drone can hover over any point in the monitored area. However, they have limited flying time due to their batteries. Hence, the algorithms for Big Data Gathering must consider both limitations of the sensor nodes and mobile collectors. Since each sensor node has a large volume of data stored in their memory, the drone must stay at least a minimum time inside each sensor node’s radio range to receive all data. Hence, we assume the drone eventually has to hover over some locations to perform the total data gathering. The majority of related works found in the literature consider sensor nodes having only a single data package to transmit. Therefore, the mobile collector does not need to stop for data gathering, and they do not consider the minimum time it must stay inside the sensor nodes’ radio range.

Silva and Nascimento [12] proposed two heuristics to define a sequence of points (hovering points) where the drone has to hover to gather data and the subset of sensor nodes that will send its data to the drone over each hovering point. They considered that sensor nodes contain a relatively large volume of data stored in their memories. The drone only receives data packets when it is hovering; there is no transmission when the drone is moving. This strategy guarantees that the mobile collector will stay a minimum time inside each sensor node’s radio range to receive all data. The proposed heuristics aim to reduce the overall time for data gathering since the drone has limited flight time.

This work proposes to improve the heuristics presented by Silva and Nascimento [12] by considering data gathering also during the drone movement. We also focus on reducing the drone data gathering time. The algorithm proposed here uses the strategies presented in the mentioned work to define the sequence of hovering points where the drone has to hover to gather data. However, our algorithm defines a subset of sensor nodes that will send data packets to the drone when traveling between each hovering point pair. We guarantee that the drone will stay at least a time long enough to receive all data packets sent from every sensor node in the path during the movement. Using this strategy, the drone reduces the hovering time and the overall time for data gathering.

The remainder of this paper is organized as follows. Next, we reviewed some related works. In Section 3, the problem is defined. Section 4 describes how to model the problem as graphs and Section 5 presents the heuristics proposed by [12]. Section 6 describes the proposed algorithm. Section 7 presents the simulation results to evaluate our proposal. Finally, Section 8 presents our conclusions and future works.

## 2. Related Works

There are several types of research about mobile collectors in WSN. Some of them focused on data gathering in large scale WSN. However, most of them only focus on network energy-saving and do not consider the mobile collector’s limitations. Furthermore, they do not consider sensor nodes storing a large volume of data to send to the mobile collector.

Wang et al. [13] present an algorithm for data collection based on neural networks. The algorithm first divides the sensor field into several equal squares to create clusters, and each cluster selects a cluster head (CH). Then, the cluster members transmit data to their corresponding CH, directly or using relay nodes. Data fusion is conducted on each CH by using a pretrained neural network when the CH receives data from all its members. Finally, the merged data is sent to the mobile collector. However, the authors do not consider the amount of data to be transmitted by each sensor or the drone battery’s limitation.

Zhang et al. [14] focused on the data gathering problem of maximizing the volume of information obtained by a mobile sink from rechargeable sensor nodes. The sensor nodes store data collected from the environment and harvest energy from the sun or wind. The mobile sink has to move on a straight road within the monitored area to gather the highest possible volume of data, considering sensor nodes with different energy levels. The authors proposed the Distributed Data Gathering Approach (DDGA), a distributed algorithm to obtain the near-optimal solution by generating data gathering routes. However, they do not control the sink movement to optimize energy consumption.

Pang et al. [15] developed a scheme for collaborative data collection using multiple mobile nodes (MN) as a sink. The randomly arranged sensor nodes are divided into clusters, and manually the cluster heads (CH) are positioned in the center of each cluster, with higher energy levels. The CHs receive and store data from all nodes in their respective clusters and wait for an MN to send data. This work studies routing strategy and path planning for multiple MNs to reduce network energy consumption. However, the authors do not consider the amount of data sent by each sensor node. Hence, they do not guarantee that the MNs will stay a minimum time inside each CH’s radio range.

Xu et al. [10] proposed the Data Quality Maximization (DQM) routing protocol to transmit data to a mobile collector. This protocol assumes the mobile sink movement’s predictability and selects as gateways the sensor nodes adjacent to the path followed by the collector. The sensor nodes use the Floyd-Warshall algorithm [16] to establish the shortest data path between each node and the closest gateway. The gateways aggregate received data and wait until the mobile is close enough to transmit. The authors consider the mobile sink moving with constant speed; hence the sink does not have time to receive a large volume of data.

Khan et al. [17] proposed an algorithm based on virtual grids to divide the WSN into clusters and define routes to deliver data to the mobile sink. Each cluster elects a node as its cluster head, which is responsible for receiving data from the other nodes in the cluster. The cluster heads together form a backbone to relay data to the mobile sink, which moves counterclockwise inside the sensor field. The uthors focused on reducing the network energy consumption by defining the data routes from the backbone to the mobile sink in constant movement.

Chen et al. [18] proposed algorithms to schedule data mules dispatching for data gathering in WSN. They proved that the problem can be solved by linear time complexity algorithms when the handling time is uniform and that the problem is NP-Complete when the handling time is nonlinear. Hence, the article proposed a linear algorithm and an approximate algorithm (with a guaranteed approximation ratio) to solve the analyzed problem. However, they focused on reducing the number of data mules to gather data and did not consider the data mule limitation.

He et al. [11] proposed a mobile sink trajectory algorithm to reduce the delay for data delivery and prolong the network lifetime. It shortens the trajectory length of the mobile sink and balances the load of rendezvous nodes. This algorithm is based on multiobjective particle swarm optimization. To shorten the mobile sink’s trajectory length, the authors designed a mechanism to select potential visiting points within communication overlapping ranges of sensor nodes. Additionally, according to the mobile sink’s trajectory characteristics, they designed an effective trajectory encoding method to generate a trajectory containing an unfixed number of visiting points. Hung et al. [19] also proposed a mobile sink trajectory algorithm. However, they focused on reducing the network energy consumption for data gathering. This algorithm is based on LEACH_C to create clusters and on Dijkstra to define the mobile sink trajectory. The authors of both works did not consider the limitation of the mobile collector. Furthermore, they consider sensor nodes with only one data packet to send.

Hou et al. [20] proposed an algorithm to define the tour followed by mobile sinks to collect data from WSNs. It also creates a virtual grid to divide the nodes into clusters, each with a cluster head. The proposed algorithm defines where the sink has to go after each data gathering. The author mentioned that one of the major problems is the mobile sink’s slow speed, which demands too much time to collect all data. They consider a sink with no trip distance limitation.

Takaishi et al. [21] also considered mobile collectors for data collection in large-scale WSN. They proposed a clustering algorithm to calculate the optimal number of cluster members to minimize energy consumption. However, again they do not care about the limitation of the battery of the mobile collector.

Finally, Ang et al. [8] proposed an analytical model to determine sensor nodes’ energy consumption in large-scale WSN using mobile collectors. They proposed a model to determine how nodes are divided into clusters to minimize energy consumption. In this work, the WSNs are disconnected, that is, they are subdivided into groups of geographically separated nodes. So, the nodes of a group do not communicate with other groups. They consider neither the mobile collector limitations nor nodes storing a large volume of data.

## 3. Problem Definition

The problem considered here consists of minimizing the overall data gathering time $T_{\mathrm{total}}$ of a drone as mobile collectors in a WSN with a large volume of data stored in its sensor nodes (Big Data). It is named here GBD (Gathering Big Data from WSNs). As input data, we consider *M* as a 2D rectangular monitored field whose area is a×l, infinite points inside it and $S=\{s_{1},s_{2},\cdots ,s_{\left|S\right|}\}$ a set of randomly distributed points in *M*, which correspond to the location of fixed sensor nodes, such as shown in Figure 1a. We also assumed each sensor node knows its own location. Moreover, each node has *m* bits of data stored in its flash memory to send to the drone. The radio range is *r* meters, and the link bandwidth is *b* bps. Consequently, each sensor node needs $\frac{m}{b}$ seconds to send the data stored in its memory to the drone.

The output or solution for the GBD problem is composed of *P*, $S_{\mathrm{subsets}}$ and $D_{\mathrm{mov}}$. *P* is a sequence of points $P=(p_{1},p_{2},\cdots ,p_{\left|P\right|})$ where the drone has to hover to gather data (hovering points). *P* defines a tour to be followed by the drone. However, the drone must start and finish its tour in the point $p_{0}=0,0$. $S_{\mathrm{subsets}}=\left\{N_{1},N_{2},\cdots ,N_{\left|P\right|}\right\}$ is composed of subsets of sensor nodes. Each subset represents the group of sensor nodes that have to send their data to the drone over a hovering point in *P*. The nodes belonging to the subset $N_{1}$ will send their data when the drone is hovering over the point $p_{1}$. Nodes in $N_{2}$ send data to the drone over $p_{2}$ and so forth. $D_{\mathrm{mov}}=\left\{d_{0↔1},d_{1↔2},d_{2↔3},\cdots ,d_{|P|↔0}\right\}$ is composed of subsets of sensor nodes that will send their data to the drone when it is moving. The sensor nodes in the subset $d_{0↔1}$ have to send data to the drone when it is flying between the points $p_{0}$ and $p_{1}$, the subset $d_{1↔2}$ has to send data to the drone traveling from $p_{1}$ to $p_{2}$, and so forth. Analyzing the output, $S_{\mathrm{subsets}}\cup D_{\mathrm{mov}}=S$ and $S_{\mathrm{subsets}}\cap D_{\mathrm{mov}}=\emptyset$.

Figure 1b presents an example of an output: $P=(p_{1},p_{2},p_{3})$, $S_{\mathrm{subsets}}=\left\{\left\{s_{1},s_{4}\right\},\left\{s_{8},s_{9}\right\},\left\{s_{6},s_{10}\right\}\right\}$ and $D_{\mathrm{mov}}=\left\{\left\{s_{2}\right\},\left\{s_{3},s_{5}\right\},\left\{s_{7}\right\},\left\{\right\}\right\}$. After receiving the sequence of points *P*, the drone begins its tour through the point $p_{0}$ = (0,0) and flies to the point $p_{1}$. During this part of the tour, the drone receives data from the sensor node in the subset $d_{0↔1}=\left\{s_{2}\right\}$. Hence, the drone hover over $p_{1}$ to gather data from the subset of nodes $N_{1}=\left\{s_{1},s_{4}\right\}$ and moves to the next point $p_{2}$. During this movement, it receives data from the sensor nodes in the subset $p_{1-2}=\left\{s_{3},s_{5}\right\}$. Then, the drone hovers over $p_{2}$ to gather data from $N_{2}=\left\{s_{8},s_{9}\right\}$ and repeats this operation until the data from all sensor nodes is collected. At the end of the tour, the drone returns to the point $p_{0}$.

We define $T_{\mathrm{total}}$ as the drone’s overall time to follow the sequence of hovering points in *P* and gather all data from all sensor nodes’ memories. This time is calculated according to Equation (Equation 1):(1)$$T_{\mathrm{total}}=T_{\mathrm{trip}}+T_{\mathrm{collecting}}$$
where $T_{\mathrm{trip}}$ is the time the drone spend moving from $p_{0}$, following each point in *P* and returning to $p_{0}$. $T_{\mathrm{trip}}$ considers only the drone movement; hence, it depends on the drone speed and the distance between each couple of hovering points in the sequence defined by *P*. $T_{\mathrm{collecting}}$ is the time the drone spends hovering over the points in *P* to collect all data from the nodes in $S_{\mathrm{subsets}}$. The following equation defines it:(2)$$T_{\mathrm{collecting}}=T_{\mathrm{allnodes}}-T_{\mathrm{moving}}$$

In Equation (Equation 2), $T_{\mathrm{moving}}$ is the time the drone spends receiving data when it is moving. In other words, it is the time needed to collect data from the sensor nodes in $D_{\mathrm{mov}}$. $T_{\mathrm{allnodes}}$ it the time to receive data from all sensor nodes without considering drone movement during the data gathering. $T_{\mathrm{allnodes}}$ is calculated by finding the shortest path between each sensor node and one of the hovering points in *P*, summing the size in hops of each path, and multiplying it by the time spent by a sensor node to send all data in its memory on one hop.

The heuristics proposed by Silva and Nascimento [12] do not consider the drone gathering data when it is moving, hence $T_{\mathrm{collection}}=T_{\mathrm{allnodes}}$. The main contribution of our work is to consider data gathering during the drone movement. The algorithm proposed here keeps the same $T_{\mathrm{trip}}$ found in the aforementioned work and reduces $T_{\mathrm{collection}}$ by $T_{\mathrm{moving}}$, according to Equation (Equation 2). Figure 2 presents an example of output created by the heuristics proposed by Silva and Nascimento [12]. Figure 1b presents an example of output created by the algorithm proposed here. We can verify that the proposed algorithm reduces the time the drone has to hover to gather data and, consequently, reduces $T_{\mathrm{total}}$. Summarizing, the problem here is how to increase $T_{\mathrm{moving}}$ in order to decrease $T_{\mathrm{total}}$.

The GBD problem belongs to the NP-hard class since it has, as a particular case, the Traveling Salesman Problem (TSP), which is NP-hard [22]. In fact, in this analogy, it is enough to consider each city as one point in *P* and the traveling salesman as the drone with sufficient capacity to collect all sensor nodes’ data. The drone must exit from $p_{0}$ and go through all the points to collect the information and return to $p_{0}$ traveling the shortest path.

In Table 1, we present the notations for this problem.

## 4. Modeling the GBD Problem

Since the monitored area is composed of an infinite set of points (*M*), we defined $P_{\mathrm{grid}}$ as a finite set of points where the drone can hover to gather data. These points are named Possible Drone Points (*PDP*), such that $P_{\mathrm{grid}}=\left\{{\mathrm{PDP}}_{1},{\mathrm{PDP}}_{2},\cdots ,{\mathrm{PDP}}_{|P_{\mathrm{grid}}|}\right\}$ and $P_{\mathrm{grid}}\subset M$. The *PDPs* form a grid inside the monitored area. The distance between two adjacent *PDPs* horizontally or vertically is $r\sqrt{2}$ meters, where *r* is the radio range of the sensor nodes (same as the drone). Figure 3 illustrates a monitored area, a set of *PDPs* forming a grid and the radio range. In this way, if a drone hovers over all *PDPs* in $P_{\mathrm{grid}}$, we guarantee that the drone can establish direct communication with all sensor nodes inside the monitored area.

The problem of gathering big data in WSNs was modeled by Silva and Nascimento [12] using two graphs, named *Trip Graph* (*TG*) and *Connectivity Graph* (*CG*).

$\mathrm{TG}=(P_{\mathrm{grid}},F)$, where $P_{\mathrm{grid}}$ is the set of all *PDPs* and *F* is a set of edges that represents the shortest distance between every pair of two *PDPs*. The Euclidean distance between two *PDPs* is the weight of each edge. $T_{\mathrm{trip}}$ is calculated by using *TG*.

The *Connectivity Graph* represents the data paths between the drone hovering over each *PDP* and every sensor node. CG=(V,E), where *V* is a set of vertices that represents the sensor nodes and the *PDPs*, such that $V=S\cup P_{\mathrm{grid}}$. *E* is a set of edges. Each edge represents the data path between the drone over a *PDP* in $P_{\mathrm{grid}}$ and a sensor node. Each vertex representing a *PDP* is connected to every vertex representing sensor nodes by an edge, which has a weight representing the distance in hops between them (One hop is a link of direct communication between two sensor nodes or between a sensor node and the drone. A path with two hops has three sensor nodes: a source, a destination, and a node between these two to relay packets.). There is no edge connecting vertex representing sensor nodes.

Figure 4a shows an example of a WSN and Figure 4b the correspondent *CG*. Figure 4a presents a WSN composed of three sensor nodes ($S=\{s_{1},s_{2},s_{3}\}$) and four *PDPs* ($P_{\mathrm{grid}}=\left\{{\mathrm{PDP}}_{1},{\mathrm{PDP}}_{2},{\mathrm{PDP}}_{3},{\mathrm{PDP}}_{4}\right\}$). Figure 4a also shows all possible links of communications between two sensor nodes and between the drone over every *PDP* and the sensor nodes. The *CG* presented by Figure 4b has weights on each edge. These weights represent the distance in hops between the drone over a *PDPs* and a sensor node. Let us take the sensor node $s_{3}$ as an example. It has three edges, two of them have weight 1. These edges represent possible direct communication between $s_{3}$ and the drone hovering over ${\mathrm{PDP}}_{3}$ and ${\mathrm{PDP}}_{4}$. The other edge of $s_{3}$ has weight 3, representing a data path with 3 hops linking $s_{3}$ and ${\mathrm{PDP}}_{1}$ ($s_{3}⟶\;s_{2}⟶\;s_{1}⟶\;{\mathrm{PDP}}_{1}$).

## 5. Heuristics for the GBD Problem

Silva and Nascimento [12] proposed two heuristics, named Incremental and Decremental, to treat the GBD problem. They create tours composed of a sequence of hovering points, where the drone hovers to gather data from the sensor nodes. The drone does not receive data when it is moving between two hovering points. Here, we propose to improve these heuristics by considering data gathering during the drone movement. The improvement proposed here starts from a tour created by any of these heuristics and seeks subsets of sensor nodes that will send data to the drone when it is moving. Hence, it is essential to describe these heuristics to understand the improvements proposed here.

The Incremental and Decremental heuristics have a construction phase to create a new solution in each loop iteration and a local search phase to improve the current solution. Both heuristics stop their loop iteration when the current solution is not better than the last one. Both follow the well-known Proximate Optimality Principle [23]. According to POP, “good solutions at one level are likely to be found close to good solutions at an adjacent level”. Here, the term level refers to a stage of the constructive process. In the following, Section 5.1 describes the Incremental Heuristic and Section 5.2 the Decremental Heuristic.

### 5.1. Incremental Heuristic

The Incremental Heuristic creates a drone tour at each loop iteration. The first tour $R_{1}$ is composed of only one *PDP*. Then, it performs the local search to improve $R_{1}$ (described in Section 5.3). In the second iteration, the heuristic adds a new PDP to $R_{1}$ in order to create the tour $R_{2}$ (composed of two *PDPs*) and performs the local search again. In other words, at each iteration a new $PDP_{j}$ is added to the tour ($R_{i+1}=R_{i}\cup PDP_{j}$), the local search is performed over this tour and $T_{\mathrm{total}}$ is calculated. The algorithm ends when the $T_{\mathrm{total}}$ value does not decrease further with the addition of a new *PDP* in the tour.

The algorithm creates a weight $w_{j}$ for each *PDP* in $P_{\mathrm{grid}}$. At each iteration, it chooses a *PDP* to add to $R_{i}$ as the *PDP* with the highest $w_{j}$. However, the chosen *PDP* must be at least 2r away from the other *PDPs* in $R_{i}$. If there is no *PDP* with this characteristic, the minimum distance will be divided by 2. Using this strategy, this algorithm avoids *PDPs* close to each other in $R_{i}$. The weight $w_{j}$ is calculated according to Equation (Equation 3):(3)$$w_{j}=\sum_{h=1}^{z}\frac{hop_{h}}{h}$$
where $hop_{h}$ is the number of sensor nodes in *CG* that are connected to the *PDP*
$p_{j}$ by edges with weight equal to *h*, and *z* is the biggest weight of the edges that link $PDP_{j}$ to the sensor nodes in *CG*. Since the weights of the edges in *CG* correspond to the distance in hops between a sensor node and the drone over $PDP_{j}$, the *PDPs* on crowded regions receive the highest weight. Figure 5a presents how to calculate the weight of each *PDP*. As an example, the drone over the $PDP_{1}$ uses one hop (direct communication) to communicate with two sensor nodes, two hops to communicate with one node, three hops to communicate with two nodes, and four hops to communicate with one sensor node. Hence, the weight of the $PDP_{1}$ is $w_{1}=\frac{2}{1}+\frac{1}{2}+\frac{2}{3}+\frac{1}{4}=3.41$.

The Incremental algorithm works according to Algorithm 1. The lines 6 and 20 are not in the original Incremental algorithm. We added them to this pseudocode to show how to use the proposed algorithm. Incremental receives as parameters the set *S* with the locations of all sensor nodes, the set of all *PDPs* in $P_{\mathrm{grid}}$, the amount of data stored in each sensor node *m*, the network bandwidth *b*, and the drone speed dspeed. In line 2, the variable that will store the best $T_{\mathrm{total}}$ is initialized with infinity. In lines 3 and 4, the set forming the best drone tour *P* and the set with the data paths between sensor nodes and each hovering point in *P* are initialized to empty. In line 5, the variable *route* also receives empty; it stores the tour created at each iteration. As aforementioned, the line 6 is not in the original Incremental algorithm. We put this line in the pseudocode to show how to use the proposed algorithm for Big Data Gathering During Drone Movement. In this line, we initialized the set of subsets of sensor nodes that have to send data to the drone when it is moving. The Connecting Graph (*CG*) and the Trip Graph (*TG*) are created in lines 7 and 8. The counter *i* is initialized with 0. It represents the amount of *PDPs* in the current solution. Line 10 initializes the set $R_{i}$ to empty, which will be increased with a new *PDP* at each loop iteration.

From line 11 to 28, there is a loop, so that each iteration creates a new solution with one more *PDP* than the last iteration. The line 12 includes in $R_{i}$ the *PDP* returned by the function NewPDP(). This function chooses the *PDP* to be added. In line 13, the counter *i* is incremented. $T_{total}$ is calculated from line 14 to line 18. In line 19, the LSearch() function is called to perform the Local Search algorithm (described in Section 5.3) in order to try to reduce the value of $T_{total}$ calculated at each iteration. When this function finds a smaller $T_{total}$, it also returns route, that is, the drone tour of the current iteration, and st as the graph with the data routes between every sensor nodes and the *PDP*. In line 20, the $T_{total}$ is recalculated but now considering the data gathering with the drone in motion. This calculation is not in the original Incremental algorithm, but it is used at this point to show how to call the proposed algorithm for Big Data Gathering During Drone Movement.

From line 21 to 27, the algorithm evaluates if the current solution has a $T_{\mathrm{total}}$ value smaller than the best solution value found so far. Otherwise, the loop is finished. The *show* function in line 29 reports *P* and $S_{\mathrm{subsets}}$ as results. On the original result, we added $D_{mov}$ that represents the result of the algorithm proposed here.

As an example, consider the sensor network presented by Figure 5a, *r* = 60 m and the distance between two *PDPs* is 84 m. Figure 5b presents the first iteration, that creates $R_{1}=\left\{{\mathrm{PDP}}_{4}\right\}$, since the ${\mathrm{PDP}}_{4}$ has the largest weight $w_{4}$ = 4.83. Figure 5c presents the second iteration, that creates $R_{2}=\left\{{\mathrm{PDP}}_{4},{\mathrm{PDP}}_{2}\right\}$, that is, ${\mathrm{PDP}}_{2}$ has been added to $R_{1}$ to form $R_{2}$. ${\mathrm{PDP}}_{1}$ and ${\mathrm{PDP}}_{5}$ have weight greater than ${\mathrm{PDP}}_{2}$ ($w_{1}$ = 3.41, $w_{5}$ = 4.33); however, they were not added to $R_{1}$ since they are at distances less than 2r.
**Algorithm 1** Incremental Heuristic.1:**procedure**Incremental($S,P_{\mathrm{grid}},m,b,\mathrm{dspeed}$)2:    besttime⟵∞3:    P⟵⊘4:    $S_{\mathrm{subsets}}⟵⊘$5:    route⟵⊘6:    $D_{\mathrm{mov}}⟵⊘$7:    $\mathrm{CG}⟵\mathrm{CreateCG}(S,P_{\mathrm{grid}})$8:    $\mathrm{TG}⟵\mathrm{CreateTG}(P_{\mathrm{grid}})$9:    i⟵010:    $R_{i}⟵⊘$11:    **while**
$i\leq |P_{\mathrm{grid}}|$
**do**12:        $R_{i+1}⟵R_{i}\cup \left\{\mathrm{NewPDP}\left(\mathrm{CG},R_{i},\mathrm{TG}\right)\right\}$13:        i⟵i+114:        $\mathrm{st}⟵\mathrm{SpanningTree}(R_{i},\mathrm{CG})$15:        $T_{\mathrm{collecting}}⟵\mathrm{CollectingTime}\left(\mathrm{st},d,b\right)$16:        $\mathrm{route}⟵\mathrm{TSP}(R_{i},\mathrm{TG})$17:        $T_{\mathrm{trip}}⟵\mathrm{TripTime}\left(\mathrm{route},\mathrm{TG},\mathrm{dspeed}\right)$18:        $T_{\mathrm{total}}⟵T_{\mathrm{collecting}}+T_{\mathrm{trip}}$19:        $T_{\mathrm{total}}⟵\mathrm{LSearch}\left(R_{i},\mathrm{CG},\mathrm{TG},d,\&\mathrm{st},\&\mathrm{route}\right)$20:        $T_{\mathrm{total}}⟵\mathrm{CreateSubsets}\left(P,S,d_{\mathrm{speed}},b,m,r,\&D_{\mathrm{mov}}\right)$21:        **if**
$T_{\mathrm{total}}<\mathrm{besttime}$
**then**22:           $\mathrm{besttime}⟵T_{\mathrm{total}}$23:           $S_{\mathrm{subsets}}⟵\mathrm{st}$24:           P⟵route25:        **else**26:           *break*27:        **end if**28:    **end while**29:    $\mathrm{show}(P,S_{\mathrm{subsets}},D_{\mathrm{mov}})$30:**end procedure**

### 5.2. Decremental Heuristic

At each iteration, the Decremental heuristic reduces by one the number of *PDPs* in the tour compared to the previous iteration and calculates $T_{\mathrm{total}}$. The loop finishes when the tour has no *PDPs* or when $T_{\mathrm{total}}$ increases if compared with the previous iteration. The first step of the Decremental heuristic is to create the tour $R_{i}$ composed of all vertices of *CG* that represents the *PDPs* and has at least one edge connected to it. The variable *i* contains the amount of PDPs in $R_{i}$. At each iteration, the algorithm removes a *PDP* from the tour $R_{i}$ to create the tour $R_{i}-1$ and calculates $T_{\mathrm{total}}$. For each $R_{i}$, the algorithm creates a graph st⊆CG, whose vertices are the *PDPs* in $R_{i}$ and all sensor nodes. For each sensor node in st, the algorithm creates an edge connecting this node to the closest PDP. The weights of these edges are the distance in number of hops from every node to the PDP. st represents the data paths of all sensor nodes and the drone hovering a PDP.

The heuristic chooses the *PDP* to be removed from $R_{i}$ based on the *impact of removal* on the value of $T_{\mathrm{collecting}}$. When $PDP_{j}$ is removed from $R_{i}$, the edges connecting it to the sensor nodes in st also have to be removed. Hence, the nodes connected to $PDP_{j}$ must be connected to other *PDPs*. Thus, st receives other edges from *CG*. The impact of removal is calculated by subtracting the sum of the new edges’ weights from the sum of the weights of the removed edges.

Algorithm 2 presents the pseudocode of this heuristic. It receives as parameters the set *S* with the locations of all sensor nodes, $P_{\mathrm{grid}}$ the set of all *PDPs*, the amount of data stored in each sensor node *m*, the links bandwidth *b*, and the drone speed dspeed.

In line 2, the variable *besttime* is initialized with infinity. It will store the best $T_{\mathrm{total}}$. In lines 3 and 4, the set forming the best drone tour P and the set with the data paths between sensor nodes and drone $S_{\mathrm{subsets}}$ are initialized to empty. In line 5, the set of subsets of sensor nodes that have to send data to the drone when it is moving is initialized. This set is not part of the original Decremental algorithm. It is used in the proposed algorithm for Big Data Gathering During Drone Movement. The Connecting Graph (*CG*) and the Trip Graph (*TG*) are created in lines 6 and 7. The SpanningTree() function is called in line 8. It creates st, which is a graph with vertices representing the PDPs in the current tour ($R_{i}$) and all sensor nodes. This graph has an edge for each sensor node, which represents the smallest data path connecting each sensor node to a PDP in $R_{i}$. In line 9, the counter *i* is initialized with 0. It represents the amount of *PDPs* in the current solution. In line 10, the set $R_{i}$ is initialized to empty. It represents the current solution at each loop iteration.

From line 11 to 16, there is a loop to create the first tour. It look at the graph st and adds to $R_{i}$ only the *PDPs* connected to at least one sensor node. At the end of this loop, $R_{i}$ has no vertex representing a PDP without edges and *i* has the amount of PDPs in $R_{i}$. The lines from 17 to 28 are the main loop of the algorithm. At each iteration, it calculates $T_{\mathrm{total}}$ for the current solution $R_{i}$ (lines 18–22), performs the Local Search to verify if $T_{\mathrm{total}}$ can be reduced (line 23), and calls CreateSubsets() (line 24) to find the sensor nodes that will send data to drone during its motion. This call is not in the original algorithm; it is used at this point to show how to use the proposed algorithm for Big Data Gathering During Drone Movement. Right after, Incremental checks and stores if $T_{\mathrm{total}}$ is the smallest one calculated so far (lines 25 to 31 and removes one *PDP* from $R_{i}$ to create the next solution $R_{i-1}$ (line 32). If $T_{\mathrm{total}}$ is greater than the best value found so far, the loop finishes in line 30. The algorithm returns the best solution found in line 35.

As an example, lets start with the graph *st* presented by Figure 6a. It has the shortest paths between each sensor node and a PDP. This graph was created from the Connecting Graph presented by Figure 4b. First, the Decremental heuristic creates the tour $R_{3}=\left\{PDP_{1},PDP_{3},PDP_{4}\right\}$, such as presented by Figure 6b. $PDP_{2}$ was not considered because it has no edge connected to it. The first loop iteration removes $PDP_{4}$ and an edge connecting this PDP to a sensor node. Another edge is included in the graph to keep the WSN connected. Since both edges have weight 1, the impact of the removal of $PDP_{4}$ is 0. Figure 6c presents the resulting graph. $PDP_{1}$ is removed in the second loop iteration. As shown in Figure 6d, $PDP_{1}$ has an impact of removal equal to 1 (the smallest) since the heuristic replaces an edge with weight 1 with another with weight 2. The heuristic stops in the next iteration. The weights of the inserted edges are obtained in the Connecting Graph presented by Figure 4b.
**Algorithm 2** Construction Phase: Decremental Heuristic.1:**procedure**Decremental($S,P_{\mathrm{grid}},m,b,\mathrm{dspeed}$)2:    besttime⟵∞3:    P⟵⊘4:    $S_{\mathrm{subsets}}⟵⊘$5:    $D_{\mathrm{mov}}⟵⊘$6:    $\mathrm{CG}⟵\mathrm{CreateCG}(S,P_{\mathrm{grid}})$7:    $\mathrm{TG}⟵\mathrm{CreateTG}{(}_{\mathrm{grid}})$8:    $\mathrm{st}⟵\mathrm{SpanningTree}(P_{\mathrm{grid}},\mathrm{CG})$9:    i⟵010:    $R_{i}⟵⊘$11:    **for each**
$v\in P_{\mathrm{grid}}$
**do**12:        **if**
v∈st
**then**13:           $R_{i}⟵R_{i}\cup \left\{v\right\}$14:           i⟵i+115:        **end if**16:    **end for**17:    **while**
$|R_{i}|>0$
**do**18:        $\mathrm{st}⟵\mathrm{SpanningTree}(R_{i},\mathrm{CG})$19:        $T_{\mathrm{collecting}}⟵\mathrm{CollectingTime}\left(\mathrm{st},d,b\right)$20:        $\mathrm{route}⟵\mathrm{TSP}(R_{i},\mathrm{TG})$21:        $T_{\mathrm{trip}}⟵\mathrm{TripTime}\left(\mathrm{route},\mathrm{TG},\mathrm{dspeed}\right)$22:        $T_{\mathrm{total}}⟵T_{\mathrm{collecting}}+T_{\mathrm{trip}}$23:        $T_{\mathrm{total}}⟵\mathrm{LSearch}\left(R_{i},\mathrm{CG},\mathrm{TG},d,\&\mathrm{st},\&\mathrm{route}\right)$24:        $T_{\mathrm{total}}⟵\mathrm{CreateSubsets}\left(P,S,d_{\mathrm{speed}},b,m,r,\&D_{\mathrm{mov}}\right)$25:        **if**
$T_{\mathrm{total}}<\mathrm{besttime}$
**then**26:           $\mathrm{besttime}⟵T_{\mathrm{total}}$27:           $S_{\mathrm{substes}}⟵st$28:           P⟵route29:        **else**30:           *break*31:        **end if**32:        $R_{i}⟵R_{i}\setminus \{\mathrm{SmallRemovalImpact}\left(R_{i},\mathrm{st}\right)$}33:        i⟵i−134:    **end while**35:    $\mathrm{show}(P,S_{\mathrm{subsets}},D_{\mathrm{mov}})$36:**end procedure**

### 5.3. Local Search

Given a tour $R_{i}$, the Local Search phase tries to decrease $T_{\mathrm{total}}$ by exchanging each *PDP* in $R_{i}$ for one of its four neighbors in the grid (up, down, left and right). The Local Search chooses the first *PDP* in $R_{i}$, replaces this *PDP* for one of its neighbors, calculates the new $T_{\mathrm{total}}$ and verifies if the new value is the smallest so far. Then, it repeats these operations with the next *PDP* in $R_{i}$. Finally, the algorithm returns the smallest $T_{\mathrm{total}}$. The *LSearch()* function, in addition of returning $T_{\mathrm{total}}$, also returns by reference the sequence of *PDPs*, forming the new drone tour (bestroute) and bestst, the graph with edges representing the new data paths. This function is called at each iteration of the Incremental and Decremental heuristics. We verify this at line 23 of Algorithm 2 and line 19 of Algorithm 1.

Figure 7 exemplifies the Local Search. In (a), the initial tour $R_{i}=\left\{PDP_{3},PDP_{5}\right\}$ is presented. In (b), the Local Search replaces $PDP_{3}$ with its neighbor $PDP_{2}$ and calculates $T_{\mathrm{total}}$. In (c), it replaces $PDP_{3}$ with $PDP_{6}$ and also calculates $T_{\mathrm{total}}$. After that, the Local Search replaces $PDP_{5}$ with each of its neighbors ($PDP_{2}$, $PDP_{4}$, $PDP_{6}$ and $PDP_{8}$) and also calculates $T_{\mathrm{total}}$ for each exchange. The Local Search returns the tour that provides the smallest $T_{\mathrm{total}}$.

## 6. Algorithm for Big Data Gathering during Drone Movement

The algorithm proposed here receives a tour created by the Incremental or Decremental heuristics and defines the sensor nodes that will send their data to the drone when it is moving. A tour created by these heuristics is a sequence of hovering points and a subset of sensor nodes for each hovering point. All nodes in a subset have to send data to the drone when it is over a hovering point. The proposed algorithm creates a subset of sensor nodes for each path between two consecutive hovering points on tour. The nodes in these subsets have to send their data to the drone during its movement. This reduces the hovering time (here named $T_{\mathrm{collecting}}$) and consequently reduces the total time to gather all data (here named $T_{\mathrm{total}}$). Furthermore, the proposed algorithm guarantees that the drone will stay a minimum time inside each sensor node’s radio range in these subsets. This time must be greater or equal to the time each node needs to transmit all data stored in its memory.

For instance, consider the example presented by the Figure 8. It shows the sensor nodes $s_{1}$ and $s_{2}$ and two consecutive hovering points $p_{1}$ and $p_{2}$ that are part of a tour *P* created by one of these heuristics. Each sensor node has *r* meter of radio range, *m* bits of data storage in its memory and link with *b* bits per second. The algorithm proposed here has to create the subset of sensor nodes $d_{1↔2}$, such that during the drone flight from $p_{1}$ to $p_{2}$, it stays inside the radio range of all nodes in $d_{1↔2}$ for a period of time greater or equal to $\frac{m}{b}$ seconds. Since we consider the drone flying straight to each hovering point in constant speed dspeed, the path between $p_{1}$ and $p_{2}$ is a line. Figure 8 shows that this line is totally out of the radio range of the sensor $s_{1}$, but part of it is inside the radio range of the sensor $s_{2}$. Considering $ds_{2}$ as the length of this part, the sensor $s_{2}$ can be part of $d_{1↔2}$ only if $ds_{2}$ is long enough to drone move over for at least $\frac{m}{b}$ seconds. Node $s_{1}$ is not in the subset $d_{1↔2}$.

The example presented here has only two hovering points and two sensor nodes. However, a tour created by the aforementioned heuristics can be composed of a sequence of several hovering points. The algorithm proposed here analyzes each sensor node for each two consecutive hovering points on tour.

### 6.1. Verifying If a Sensor Node Can Send Data during the Drone Flight

This subsection presents how to verify if a sensor node $s_{i}$ can send data to a drone flying over the path between the hovering points $p_{a}$ and $p_{b}$, as exemplified by Figure 9. We consider that the area covered by a sensor node’s radio is a circle with radius *r* and center in the sensor node position. The time an object takes to travel a given distance at constant speed is the distance divided by the speed. Hence, we define trs as the minimal distance the drone has to fly inside the area covered by a sensor node to enable it to send data during the drone movement. The length of trs depends on the drone speed ($d_{\mathrm{speed}}$), the link bandwidth (*m*), and the amount of data stored in the memory of each sensor node. Consequently, trs is calculated with the following equation:(4)$$trs=\frac{m}{b}\times d_{\mathrm{speed}}$$

Let $ds_{i}$ be the part of the path $p_{a}$-$p_{b}$ covered by the radio of the sensor node $s_{i}$. The length of $ds_{i}$ is easily calculated by the following equation:(5)$$ds_{i}=2\sqrt{r^{2}-{\left(dist_{i}\right)}^{2}}$$
where $dist_{i}$ is the Euclidean Distance of the sensor node $s_{i}$ to the line $p_{a}-p_{b}$. Equation (Equation 5) is a manipulation of the Pythagorean theorem, with a=r, $b=dist_{i}$ and $c=\frac{ds_{i}}{2}$. We can see these variables in Figure 9.

The sensor node $s_{i}$ can send data to drone flying over the path between $p_{a}$ and $p_{b}$ only if $ds_{i}\geq trs$. Given a tour composed of a sequence of hovering points, the algorithm proposed here calculates $ds_{i}$ for every sensor node for each path between two hovering points and compares them to trs. Hence, the algorithm creates a subset of sensor nodes for each path and guarantees that the drone will stay a minimum time inside the sensor nodes’ radio range. The algorithm is described in the following subsection.

### 6.2. The Algorithm to Create the Subsets

A tour *P* created by Incremental or Decremental heuristic is a sequence of hovering points. We named as a *path*, part of the tour between two consecutive hovering points. Each tour has |P|+2 paths among its hovering points since the drone always starts flying from the initial point $p_{0}=0,0$ and always returns to the initial point. Consequently, the proposed algorithm creates |P|+2 subsets of sensor nodes. It uses the Equations (Equation 4) and (Equation 5) to verify if each sensor node is in every subset. Algorithm 3 presents the pseudocode to define how the proposed algorithm works.
**Algorithm 3** Gathering Big Data During Drone Movement.1:**procedure**CreateSubsets(*P*,*S*,$d_{\mathrm{speed}}$,*b*,*m*,*r*)2:    $\mathrm{trs}=MinDist(d_{\mathrm{speed}},b,m,r)$3:    $\mathrm{Paths}⟵\left\{p_{0}\right\}+P+\left\{p_{0}\right\}$4:    CG⟵SpanningTree(P,S)5:    $D_{mov}⟵⊘$6:    $T_{moving}\leftarrow 0$7:    i=08:    **for**
i<|Paths|
**do**9:        $d_{\left(i↔i+1\right)}⟵⊘$10:        u←011:        **for**
u<|S|−1
**do**12:           **if**
$DS(Paths_{i},Paths_{i+i},s_{u},r)\geq trs$
**then**13:               **if**
$s_{u}\notin D_{\mathrm{mov}}$
**then**14:                   $d_{\left(i↔i+1\right)}⟵s_{u}$15:               **end if**16:           **end if**17:        **end for**18:        $D_{mov}⟵D_{mov}\cup OverlappingNodes\left(d_{\left(i↔i+1\right)},CG\right)$19:    **end for**20:    $T_{moving}\leftarrow CalcTmov\left(D_{mov},CG\right)$21:    $show(D_{mov},T_{Moving})$22:**end procedure**

Algorithm 3 receives as parameters a tour *P*, the set *S* with the locations of all sensor nodes, the drone speed dspeed, the network bandwidth *b*, the amount of data stored in the memory of each sensor node *m*, and the sensor nodes radio range *r*, the same as the drone. The line 2 calculates a threshold (trs) that is the minimal distance the drone has to fly inside the area covered by the radio of a sensor node to receive data from it. The function MinDist() works according to Equation (Equation 5). In line 3, the variable *Paths* receives the tour *P* with the initial point $p_{0}$ at the beginning and at the end, in order to represent the entire drone trip. The Connecting Graph is created by the SpanningTree() function in line 4. $D_{\mathrm{mov}}$ is initialized in line 5. It is the set of subsets of nodes that send data to drone when it is moving. The variables $T_{\mathrm{moving}}$ and *i* are initialized in lines 6 and 7, respectively.

From line 8 to 19, there is a loop that takes separately each pair of consecutive hovering points in *Paths*. From line 11 to 17, the loop takes separately each sensor nodes. In line 12, it checks if the drone will fly at least trs meters inside the radio range of the sensor node. In line 13, it verifies if this sensor node already is in one of the subsets of $D_{\mathrm{mov}}$. If not, in line 14, this sensor node is added to the set that will send data to the drone between the points *Paths* and ${\mathrm{Paths}}_{i+1}$. In line 18, the algorithm searches for sensor nodes that are in the same coverage region. As only one sensor node can send data at a time, the procedure *OverlappingNodes*() checks which sensor node will send data in each region of coverage. We describe this procedure in Section 6.2.1. In line 20, it calculates $T_{\mathrm{moving}}$, which is the time that will be removed from the collection time ($T_{\mathrm{collecting}}$), according to Equation (Equation 2). The Section 6.2.2 describes how to calculate $T_{\mathrm{moving}}$. The result is shown in line 21.

#### 6.2.1. Check for Overlapping

Given a path between two consecutive hovering points in a tour, it is possible that during the drone flight over this path, the drone will be inside of areas covered by more than one sensor node at the same time. In other words, during the drone flight, it can cross regions with *overlapping* of radio ranges. To avoid packet collisions in this scenario, only one node at a time can send data to a drone. Figure 10a exemplifies this overlapping. It shows a path between the hovering points $p_{i}$ and $p_{i+i}$, the sensor nodes $s_{1}$ and $s_{2}$, and their radio ranges. Since these nodes are close to each other, the area covered by their radios on the tour is almost the same.

The proposed algorithm avoids overlapping by defining how the sensor nodes transmit to the drone and remove some nodes when the overlapped region is not large enough for the drone to receive data from all sensor nodes. Lets consider $d_{i↔i+1}$ the subset of sensor nodes able to send data to drone flying on the path between $p_{i}$ and $p_{i+1}$. First, the algorithm sorts the nodes in $d_{i↔i+1}$ according to the sequence of nodes perceived by the drone during its flight between $p_{i}$ and $p_{i+1}$. On this path, the algorithm defines where each sensor node can communicate with the drone. Then, it allocates part of the path $p_{i}-p_{i+1}$ to the first sensor node in $d_{i↔i+1}$. This part starts where the drone enters inside of the radio range of this node and has length trs, according to Equation (Equation 4). Hence, it verifies if the next node in $d_{i↔i+1}$ can transmit data after the first node, that is, after the end of the first part. If so, the algorithm allocates this second part to the second node. If not, it verifies the next sensor node. This task is repeated until the last sensor node in $d_{i↔i+1}$. Then, the algorithm verifies the next subset in $D_{\mathrm{mov}}$. Figure 10b presents the path $p_{i}-p_{i+1}$ with the first part allocated to sensor node $s_{1}$ and the second part allocated to sensor node $s_{2}$.

#### 6.2.2. Calculating $T_{\mathrm{moving}}$

The Incremental and Decremental heuristics consider data gathering only when the drone is hovering. Hence, the time the drone spends hovering ($T_{\mathrm{collecting}}$) is the time all nodes need to send data to a drone ($T_{\mathrm{allnodes}}$). It considers multihop data paths from sensor nodes to the drone. The overall time to gather all data from the WSN ($T_{\mathrm{total}}$) is the sum of $T_{\mathrm{collecting}}$ with the time the drone spends flying to reach each hovering point ($T_{\mathrm{trip}}$), according to Equation (Equation 1).

The proposed algorithm has a procedure to create the set $D_{\mathrm{mov}}$ composed of a subset of sensor nodes for each path between two consecutive hovering points on tour. $T_{\mathrm{moving}}$ is the time the drone spends receiving data during its movement. It reduces $T_{\mathrm{collecting}}$ since it decreases the number of sensor nodes to send data when the drone is hovering. This is the main contribution of this work since the state-of-the-art heuristics for the scenario considered here does not support data gathering when the sink is moving. The reduction in $T_{total}$ can be noted by comparing the same heuristic to create tours with and without the proposed algorithm. These results are explored in the next section. The procedure described here calculates $T_{\mathrm{moving}}$ according to Equation (Equation 6). It sums the time each sensor node in $D_{\mathrm{mov}}$ would spend to send data to the drone when it is hovering. The Connecting Graph, defined in Section 4, is used to verify the data paths’ length.
(6)$$T_{\mathrm{moving}}=\sum_{k=1}^{|D_{\mathrm{mov}}|}\frac{m}{b}\times pathlength\left(s_{k}\right)$$
where $|D_{\mathrm{mov}}|$ is the cardinality of the set $D_{\mathrm{mov}}$, *m* is the amount of data storage in the memory of each sensor node, *b* is the link bandwidth, and $pathlength(s_{k})$ is a function that is obtained from the Connect Graph, the data path length of the sensor node $s_{k}$.

## 7. Experiments

We create simulated experiments to evaluate the proposed algorithm’s performance for big data gathering in WSN during the mobile collector movement. We have implemented the Incremental and Decremental heuristics proposed by Silva and Nascimento [12]. The only change made on these heuristics was the algorithm for solving the TSP. We included the Concorde Solver [24], a state-of-the-art algorithm to solve the TSP, to reduce the execution time of these heuristic methods. The original heuristics used a brute force algorithm that increased the execution time for larger monitored areas too much. It is important to mention that the metric execution time is not analyzed here. Both the Concorde and the brute force TSP solver provide the same results since they are exact algorithms. However, Concorde was used to reduce the time we executed our simulation. Then, we applied our algorithm on each tour created by every loop iteration of these heuristics (after lines 19 and 23 of the Algorithms 1 and 2, respectively) and saved the best solution. Hence, the graphs presented in the following show four heuristic methods: Incremental and Decremental, representing the original heuristics, and Incremental-Move and Decremental-Move representing the heuristics with the proposed algorithm. We implemented all methods in Java 8–64 bits. The computer used to run the experiments has a processor Intel Core I7 8565U 1.8GHz and 8 GB of RAM.

The experiments were performed in the same scenarios and used the same characteristics described in [12], where the Incremental and Decremental heuristics were described. We divided these experiments into two phases. In the first phase, we defined the Scenario 1, that is a small square monitored area with 200 m of side and 30 fixed sensor nodes. In the second phase, we consider a larger monitored area and a larger number of sensor nodes to evaluate the heuristics’ performance in WSN with data routes longer than in the first scenario. This phase has Scenarios 2, 3, and 4. All of them have a square monitored area with 400 m of side. Scenarios varied the amount of data stored in each sensor node memory from 20 to 120 kbits. Scenario 3 varied the number of sensor nodes from 100 to 250. Finally, Scenario 4 varied the drone speed from 0.5 to 3.0 m/s. Table 2 summarizes all characteristics of the four scenarios.

We consider a drone with hovering capability as the mobile sink, such as a quadcopter. It is able to fly and hover over any point in the monitored area. The mobile sink has a radio like the sensor nodes, with the same range of the nodes (r = 60 m). The transmission rate of every link the WSN is 20 kbps. In all scenarios, $P_{grid}$ is composed of *PDPs* with 84 m of distance between two of them (up, down, left, and right). In this way, every point inside the monitored area is less than 60 m far from a *PDP*. The time to propagate queries is not considered here. The drone moves at a constant speed and collects data when it is hovering and when it is moving. The main metric is $T_{\mathrm{total}}$. Every point plotted on the graphs represents the average of 33 simulations using different WSN topologies, which provides a 95% confidence interval.

In Section 7.1 and Section 7.2, we compare the performance of the Incremental, Decremental, Incremental-Move, and Decremental-Move methods in the small and large monitored areas, respectively. In Section 7.3, we verify if there is a statistical difference among the methods evaluated here.

### 7.1. Small Monitored Area

Here, we consider Scenario 1 that is composed of 30 sensor nodes uniformly randomly deployed on a 2D square monitored area with 200 m of side. The drone flies at a constant speed of 2 m/s and hovers over the tour’s hovering points. In the graph of Figure 11, we increased the amount of data stored in each sensor node and plotted these values on the X-axis. The Y-axis represents $T_{\mathrm{total}}$ obtained by the four heuristic methods. We verify that $T_{\mathrm{total}}$ increases when the amount of data in the sensor nodes increases, as expected. Incremental-Move outperformed Incremental in all scenarios, mainly in scenarios where the sensor nodes have more data to transmit. The same happens with Decremental-Move and Decremental. This shows that the proposed algorithm can effectively find sensor nodes able to send data to the drone in movement and, consequently, reduce the hovering time. Comparing only the heuristics that received the proposed algorithm, we verify that in scenarios when nodes are storing a smaller amount of data, Incremental-Move outperformed Decremental-Moved. However, in scenarios when nodes store a larger amount of data, Decremental-Moved presents the best results. This happens because the Decremental heuristic tends to find tours with more hovering points than the Incremental heuristic. Hence, the drone’s path tends to be larger, and the number of sensor nodes able to send data during the drone movement increases.

The graphs of Figure 12 and Figure 13 help us to understand the behaviors presented by the methods in Figure 11. The time for data gathering ($T_{\mathrm{total}}$) is the sum of the times spent by the drone flying to reach each hovering point ($T_{\mathrm{trip}}$) and the time it hovers to gather data ($T_{\mathrm{collecting}}$), according to Equation (Equation 1). Figure 12 shows $T_{\mathrm{trip}}$ only for Incremental and Decremental because Incremental-Move and Decremental-Move do not change the path created by these heuristics and followed by the drone. This graph shows that when increasing the amount of data stored in each sensor node memory the tour also increases, that is, the number of hovering points in the tour also increases. Figure 13 shows that $T_{\mathrm{collecting}}$ grows when the amount of data in the sensor node memories increases. However, the growth of Incremental-Move and Decremental-Move is smaller than the growth of Incremental and Decremental. This is because the heuristics create longer tours when the nodes have more data to transmit, and the longer the tour is the larger will be the number of sensor nodes able to send data during the drone movement. Since the drone will receive data from more sensor nodes during its flight, it will reduce $T_{\mathrm{collecting}}$, consequently also reducing $T_{\mathrm{total}}$.

These experiments show that the proposed algorithm for gathering data during the drone movement (Algorithm 3) can effectively reduce the overall data gathering time. Furthermore, it presents a better performance when the tour is longer and when the sensor nodes have more data to send to the drone.

### 7.2. Larger Monitored Area

The larger monitored area considered here is a 2D square with 400 m of side. The graphs presented here show $T_{\mathrm{total}}$ of the four heuristic methods, in three different scenarios.

In Scenario 2, whose results are illustrated in Figure 14, we consider 150 uniformly randomly deployed sensor nodes and the drone’s speed of 2 m/s. We increased the amount of data stored in each sensor node (X-axis) and analyzed $T_{\mathrm{total}}$ (Y-axis). Incremental-Move and Decremental-Move presented the best performance in practically all experiments. The strategy to gather data during the drone movement reduced the overall time up to 25% when the nodes had more data stored than the original heuristics. In this scenario, the data routes tend to be longer than in the previous scenario. The tour created by the original heuristics considers several nodes to send data to the drone by data routes with many hops. Since some of these nodes send data to the drone in movement, the reduction of $T_{\mathrm{total}}$ is bigger. Here, we also verify that Decremental-Move outperformed all heuristics. This occurs because the Decremental heuristic tends to create a longer tour, increasing the number of nodes able to send data during the drone movement.

Figure 15 presents the results of the experiments in Scenario 3. In this figure, the X-axis represents the number of sensor nodes varying from 100 to 250, and the Y-axis represents $T_{\mathrm{total}}$. We consider 60 Kb of data stored in each sensor node, and the drone speed is 2 m/s. The two heuristics with the proposed Algorithm 3 outperformed the original heuristics in practically all experiments. It is important to note that the growth of all heuristics is linear. Hence, the heuristics analyzed here can be used in scenarios with more sensor nodes.

In Scenario 4, we analyze the influence of the drone’s speed on the overall data gathering time. Figure 16 presents the results of the experiments in this scenario. We consider 60 Kbits of data in each sensor node, and the number of nodes in the monitored area is 150. In this scenario, all heuristics take advantage of the higher drone speed. Even Incremental-Move and Decremental-Move, which consider data gathering during the drone movement, reduced $T_{\mathrm{total}}$ with the speed growth.

### 7.3. Statistical Analysis

We perform experiments to verify if the results obtained by the heuristics using the proposed algorithm are statistically better than the results obtained by the original heuristics proposed by [12].

We performed the paired *t*-test on the data of each graph presented in the previous section to verify if the means of $T_{\mathrm{total}}$ obtained by Incremental-Move are statistically smaller than the means obtained by Incremental and the same with Decremental-Move and Decremental. The *t*-test is a type of statistical test used to compare the means of two groups of values and evaluate if they are significantly different from each other [25]. The *t*-tests can be divided into two types: independent and paired. The independent *t*-test is used when the two groups under comparison have no relation to each other. The paired *t*-test is used when the two groups under comparison are dependent on each other [26]. In our analyses, we used the paired *t*-test because both individuals in each pair used the same heuristic to create the tour and the same network topologies.

Since the *t*-test can be used only to analyze samples with normal distribution, we first applied the Shapiro-Wilk test [27] on each sample that generated a mean to plot in the graphs. All samples presented normal distribution. Then, we applied the *t*-test on each pair of samples with the same heuristic and the same value in the X-axis. For example, in the Scenario 1 (Section 7.1), we calculated all the values of the t observed for Incremental vs. Incremental-Move, when X= 20, 40, 60, 80, 100, and 120 Kbps. The same was made for Decremental vs. Decremental-Move. These values are in Table 3. The values of the observed t for the Scenarios 2, 3, and 4 (Section 7.2) are presented in Table 4, Table 5 and Table 6, respectively. However, in Scenario 1, we analyzed only the samples plotted in the Figure 11. It is because this graphs presents the main metric ($T_{\mathrm{total}}$). The other graphs in this subsection were created only to explain the curves’ behavior in the first graph.

The number of degrees of freedom of these experiments is N−1=29, where *N* is the number of executions with different network topologies. We set the significance level to 0.001 (0.1%). The critical *t*, obtained from the table, is 3.659.

In all scenarios, we can verify that the observed value *t* is greater than the critical value *t*. Hence, we can affirm with 99.9% confidence that the proposed algorithm reduces the overall data gathering time substantially.

## 8. Conclusions and Future Works

This work analyzed the problem of finding the best drone tour plan for big data gathering in WSN. We considered the drone a quad-copter with hovering capability as a mobile sink, flying and hovering over all the monitored areas. However, it has a flying time limited by its battery. We also considered sensor nodes storing a relatively large volume of data to be gathered by the drone. Hence, the drone needs time to receive all data packets from every sensor node. We focused on finding the drone’s tour plan with the shortest time to gather data from all sensor nodes.

The state-of-the-art methods for this scenario are from Silva and Nascimento [12]. They proposed two heuristics to define drone tours to reduce the data gathering time. A tour is a sequence of locations, or hovering points, inside the monitored area. Each hovering point has a subset of sensor nodes that will send data to the drone when it is hovering over this point. The drone has to follow the sequence and hover over each hovering point for data gathering. Here, we proposed a new algorithm that receives a tour defined by one of these heuristics and creates a subset of nodes that will send data to the drone during its movement. The proposed algorithm guarantees that the drone will stay inside each sensor node’s radio range for a minimum time to receive all data. It also defines nodes’ sequence to send data to avoid two or more nodes sending data simultaneously.

Our simulated experiments showed that the proposed algorithm reduces up to 30% of the overall data gathering. Since the heuristics mentioned above create a better tour in each loop iteration, we applied the proposed algorithm in each tour. Then, we saved that one with the shortest overall data gathering time. We verified that the proposed algorithm provides better results for longer tours and when the sensor nodes have more data to transmit to a drone. We performed the *t*-test to affirm, with 99.9% confidence, that the heuristics’ results using the proposed algorithm are statistically better than those obtained from the original heuristics by Silva and Nascimento [12].

As future work, we intend to consider other sets of PDPs. The possible drone positions, or PDPs, are locations inside the monitored area where the drone can hover. These PDPs form a fixed grid, and the tour is a subset of these PDPs. We intend to vary the original PDPs to find better locations to drone gather data. Furthermore, we intend to develop other strategies to create tours. 

## Figures and Tables

**Figure 1 sensors-20-06954-f001:**
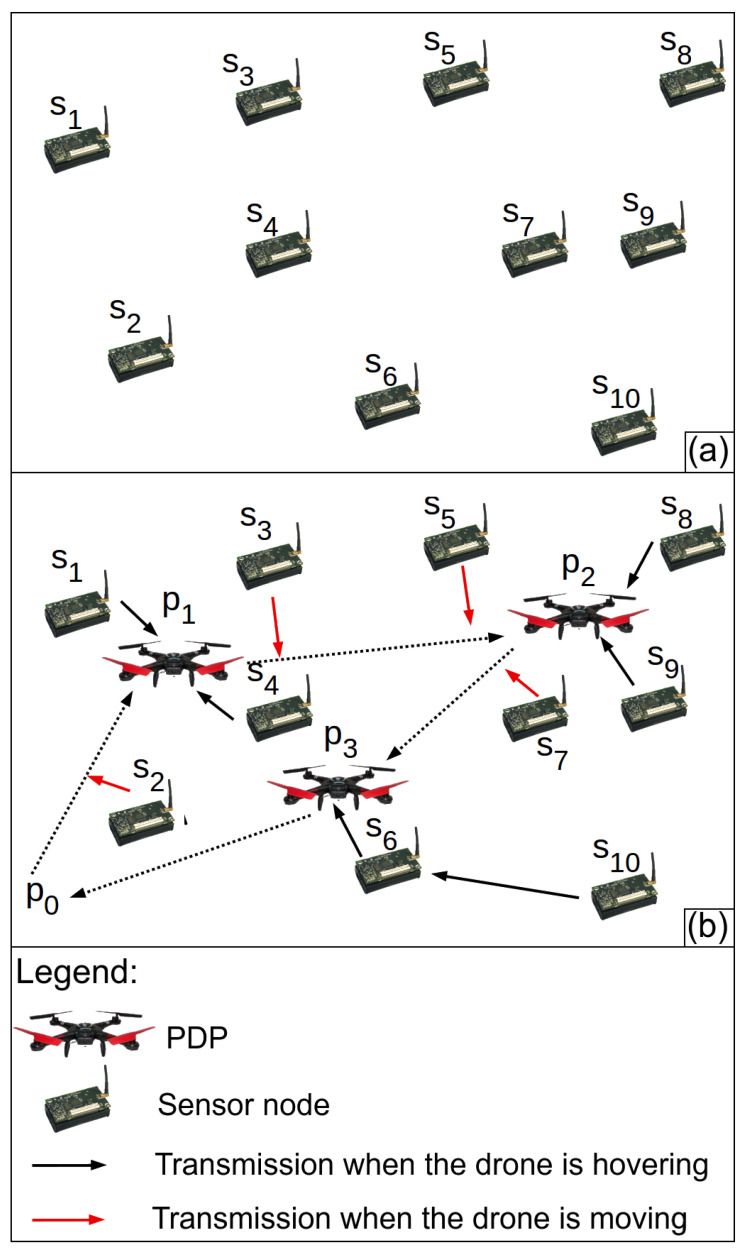
(**a**) a set of sensor nodes. (**b**) Example of output: a sequence of hovering points where the drone has to hover, a subset of sensor nodes to send data to the drone over each hovering point, and a subset of sensor nodes to send data to the drone during its movement between each pair of hovering points.

**Figure 2 sensors-20-06954-f002:**
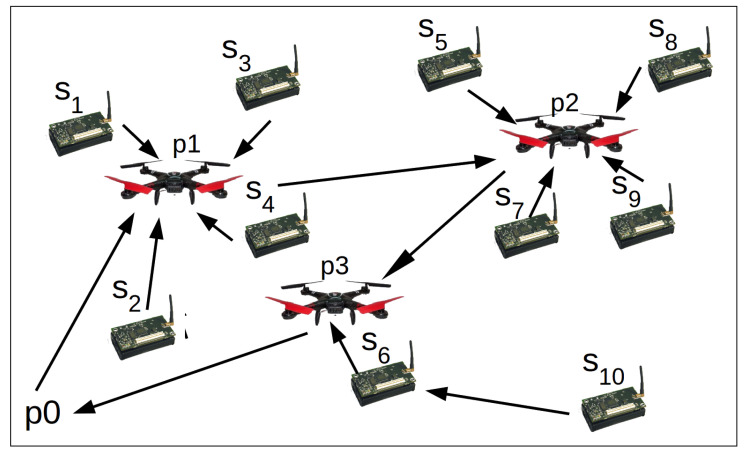
Example of output created by Silva and Nascimento [12].

**Figure 3 sensors-20-06954-f003:**
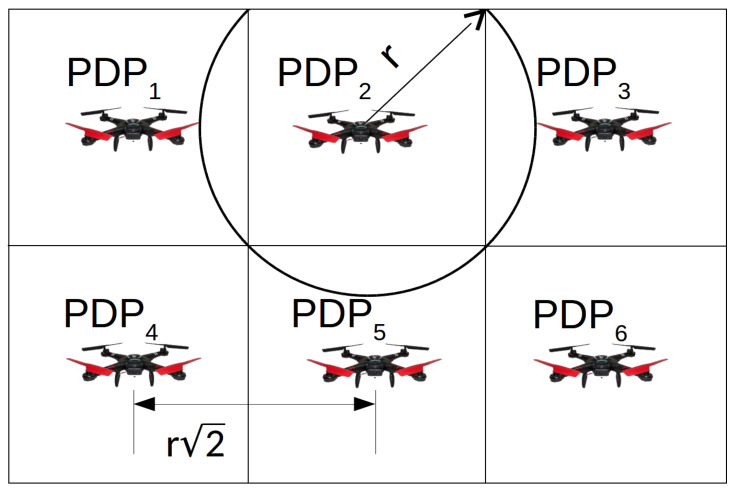
Finite set of points, named Possible Drone Points (*PDPs*), forming a grid. The drone can hover over these points for data gathering.

**Figure 4 sensors-20-06954-f004:**
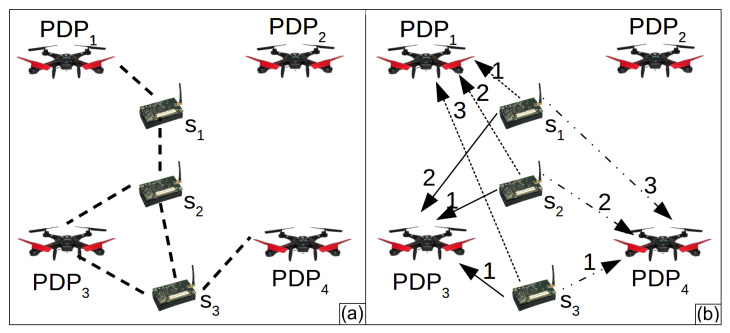
(**a**) Example of Wireless Sensor Network (WSN) and (**b**) correspondent Connectivity Graph (CG).

**Figure 5 sensors-20-06954-f005:**
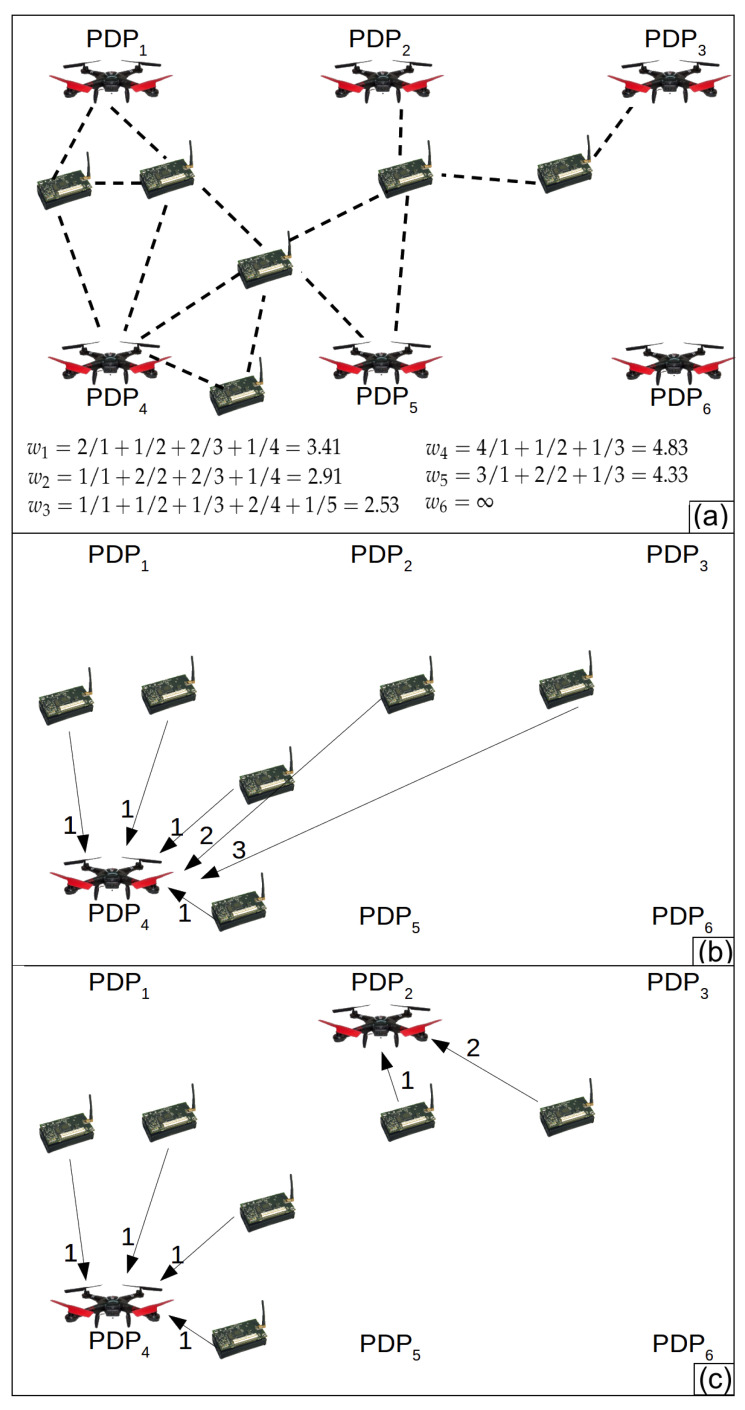
Example of how the Incremental Algorithm works. (**a**) Sensor nodes, PDPs and possible direct communication between two sensor nodes and between sensor nodes and the drone hovering each PDP. It also shows how to calculate the weights of each PDP. (**b**) First tour ($R_{1}=\left\{PDP_{4}\right\}$) created at the first loop iteration. $PDP_{4}$ has the biggest weight. (**c)** Second tour ($R_{2}=\left\{PDP_{2},PDP_{4}\right\}$) created at the second loop interaction. $PDP_{1}$ and $PDP_{5}$ were not included to $R_{2}$ because they are less then 2*r* from $PDP_{1}$.

**Figure 6 sensors-20-06954-f006:**
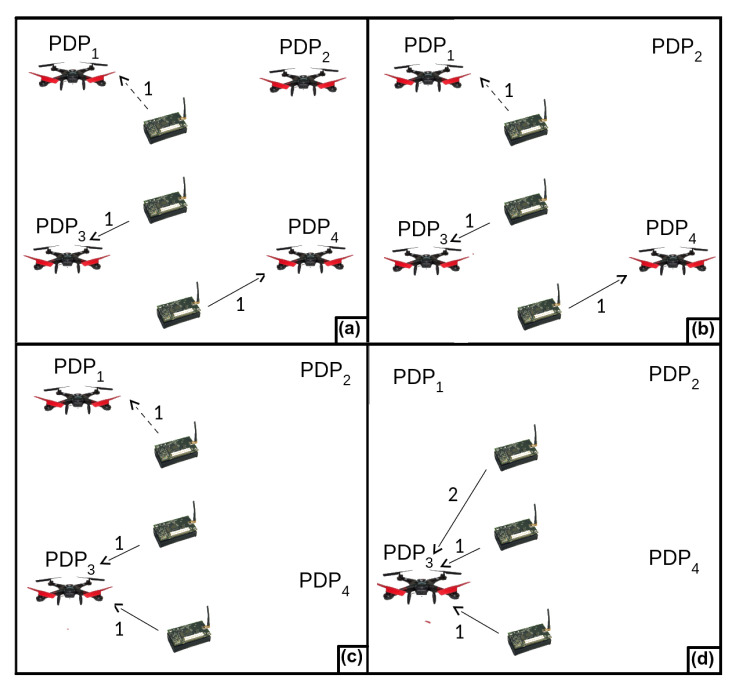
Example of the Decremental Heuristic. (**a**) The graph representing the shortest path (in hops) between each sensor node and a PDP. (**b**) The first tour $R_{3}=\left\{PDP_{1},PDP_{3},PDP_{4}\right\}$. $PDP_{2}$ was removed because it has no edge connecting it to a sensor node. (**c**) The second tour $R_{2}=\left\{PDP_{1},PDP_{3}\right\}$. $PDP_{4}$ was removed because its impact removal is the smallest, i.e., 0. (**d**) The third tour $R_{1}=\left\{PDP_{1}\right\}$. $PDP_{3}$ was removed because its impact of removal is the smallest, i.e., 1.

**Figure 7 sensors-20-06954-f007:**
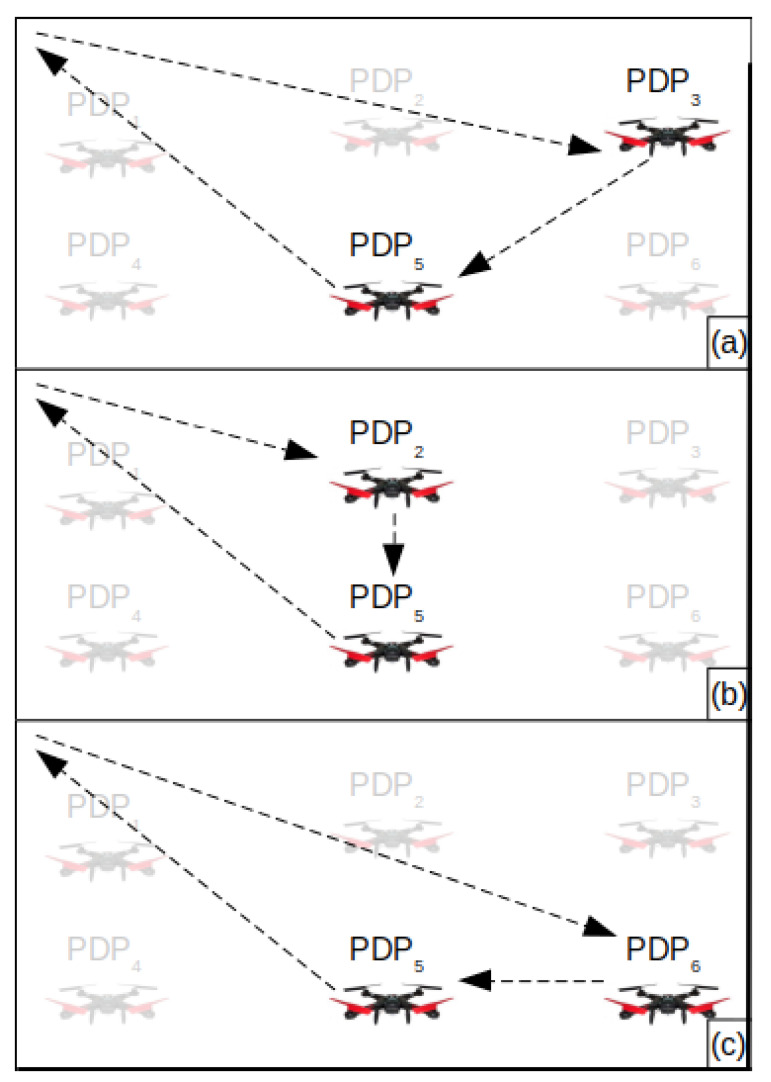
Example of the Local Search. (**a**) The initial tour $R_{i}=\left\{PDP_{3},PDP_{5}\right\}$. (**b**) $PDP_{3}$ was replaced by $PDP_{2}$ and $T_{\mathrm{total}}$ is calculated. (**c**) $PDP_{3}$ was replaced by $PDP_{6}$ and $T_{\mathrm{total}}$ is calculated again. After that, the Local Search replaces $PDP_{5}$ with $PDP_{2}$, $PDP_{4}$ and $PDP_{6}$; however, the given example does not present it.

**Figure 8 sensors-20-06954-f008:**
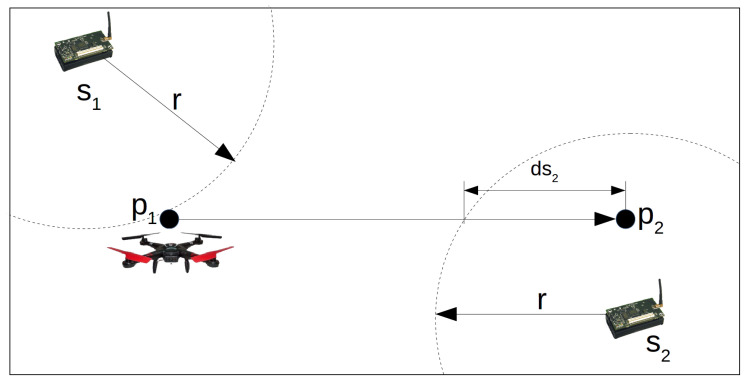
Analyzing nodes to send data to the drone during movement.

**Figure 9 sensors-20-06954-f009:**
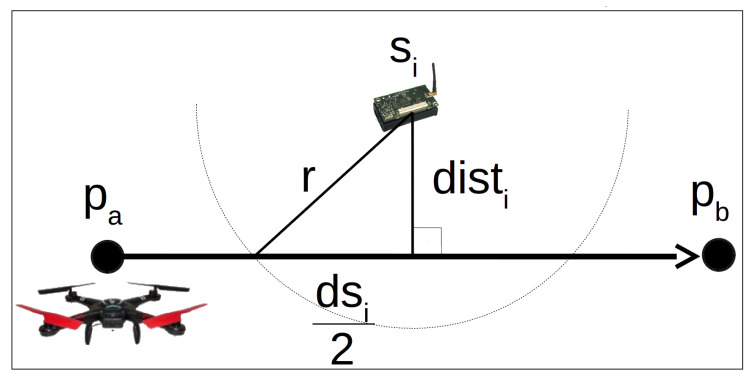
Verifying if a node $s_{i}$ can send data to drone flying between the hovering points $p_{1}$ and $p_{2}$.

**Figure 10 sensors-20-06954-f010:**
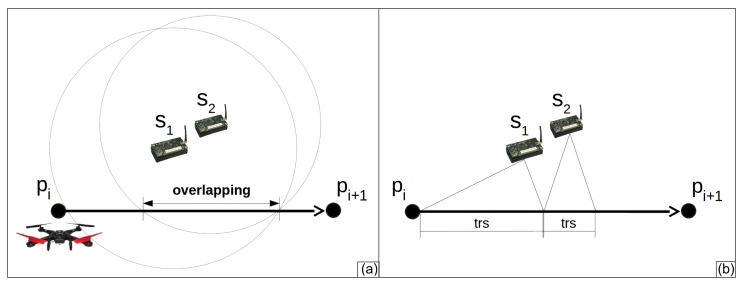
(**a**) Example of overlapping of radio ranges. (**b**) Part of the path where each node has to send data to the drone.

**Figure 11 sensors-20-06954-f011:**
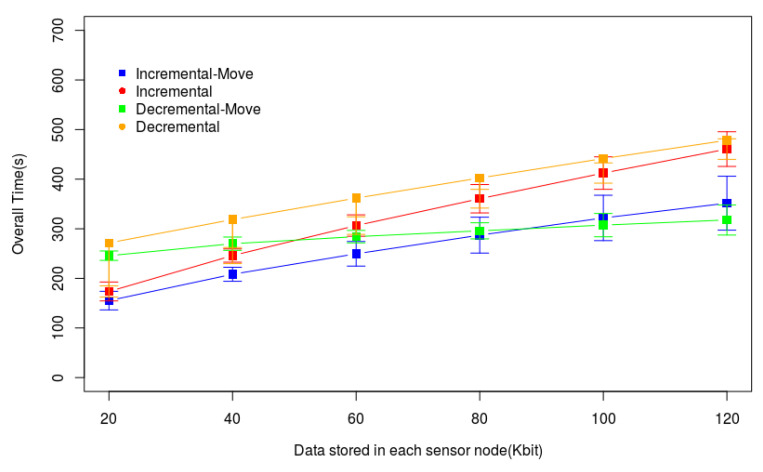
Overall data gathering time ($T_{\mathrm{total}}$) in a small monitored area.

**Figure 12 sensors-20-06954-f012:**
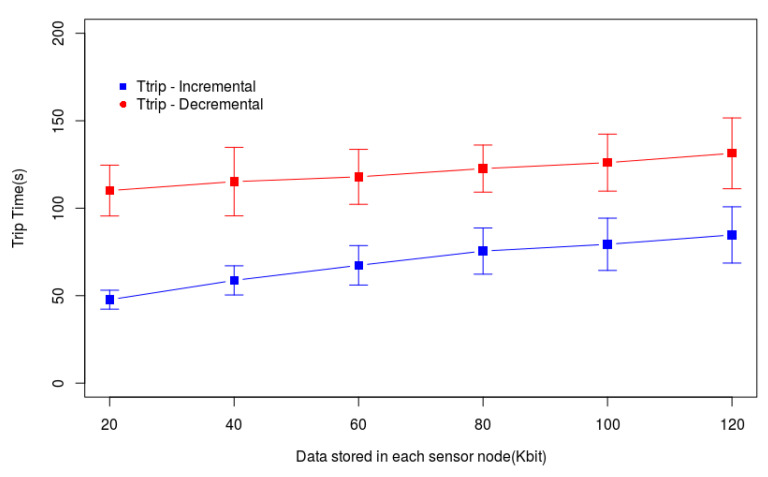
Drone moving time ($T_{\mathrm{trip}}$) in a small Monitored Area.

**Figure 13 sensors-20-06954-f013:**
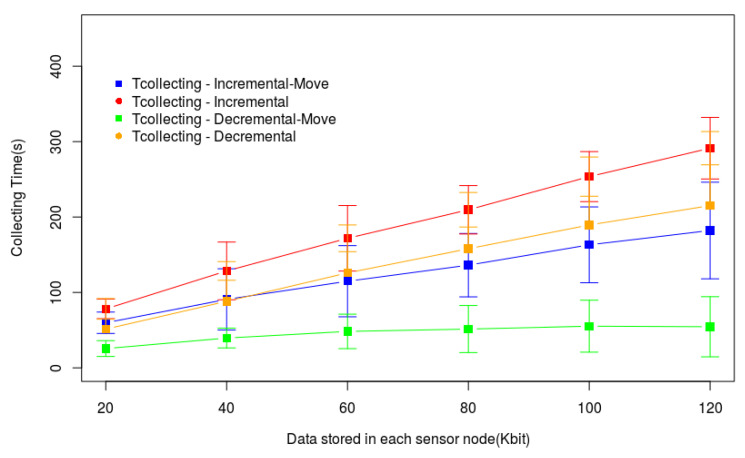
Drone hovering time for data gathering ($T_{\mathrm{collecting}}$) in a small monitored area.

**Figure 14 sensors-20-06954-f014:**
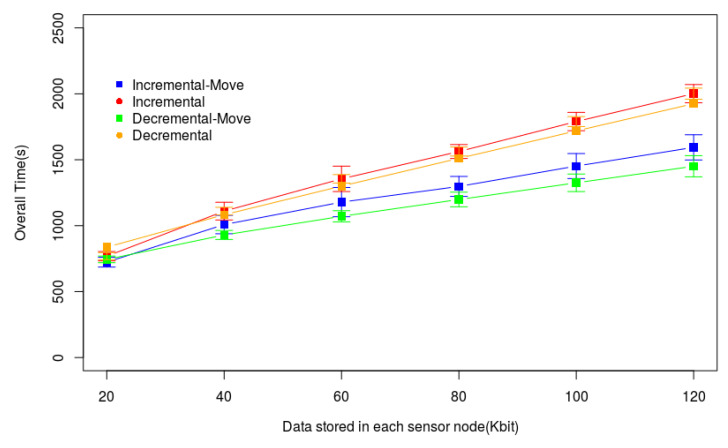
Overall data gathering time ($T_{\mathrm{total}}$) in a large monitored area.

**Figure 15 sensors-20-06954-f015:**
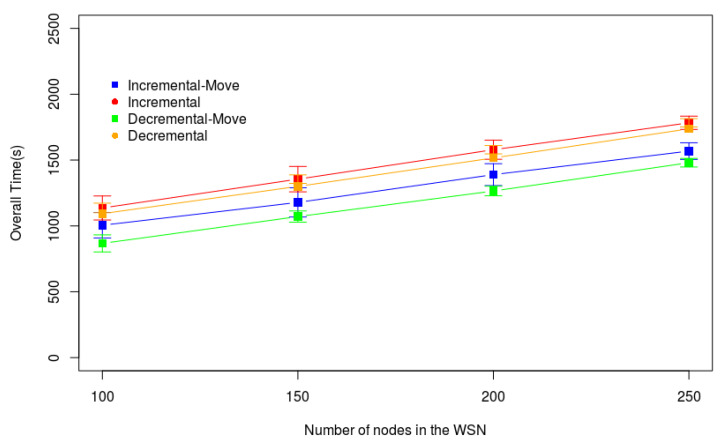
Varying the number of nodes in the WSN—Large Monitored Area.

**Figure 16 sensors-20-06954-f016:**
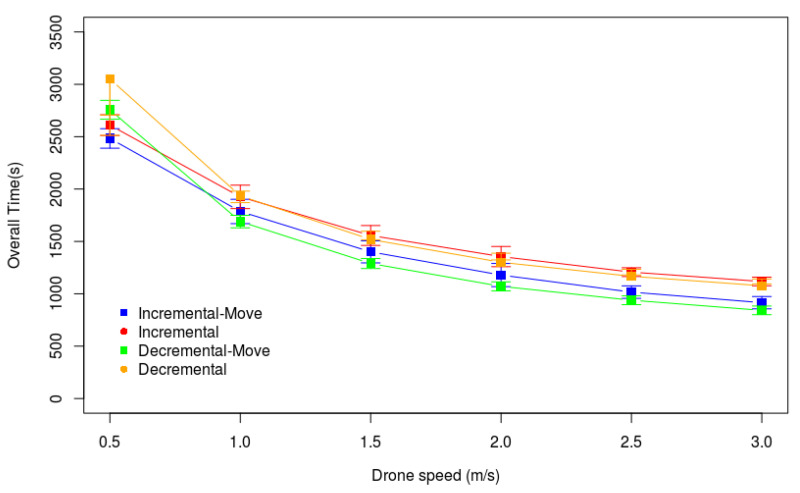
Varying the drone speed (m/s)—Larger Monitored Area.

**Table 1 sensors-20-06954-t001:** Problem Notation.

Notation	Description
*b*	The bandwidth of a link between sensor nodes or between a drone and a sensor node
CG	Connecting Graph, CG=(V,E) such that *V* is the set of vertices representing PDPs and the sensor nodes, and *E* is the set of edges representing communication links between the sensor nodes and the drone over a PDP
$d_{i↔i+1}$	Subset of sensor nodes that send data to the drone when it is flying between two consecutive hovering points $p_{i}$ and $p_{i+1}$ in a tour
$D_{\mathrm{mov}}$	Set of subsets of sensor nodes that have to send data to the drone when it is moving
$dist_{i}$	The Euclidean Distance of the sensor node $s_{i}$ to the path (line) between two consecutive hovering points in a tour
*E*	Set of edges representing communication links between the sensor nodes and the drone over a PDP in the Connecting Graph
*F*	Set of edges representing the shortest distance between every pair of *PDPs* in the Trip Graph
*m*	Amount of data stored in the memory of each sensor node
*M*	2D rectangular monitored area
$N_{j}$	Subset of sensor nodes that have to send data to the drone when it is over the hovering point $p_{j}$
$p_{0}$	Initial and final point in the drone tour, $p_{0}=0,0$
$p_{j}$	Hovering point in the drone tour, $p_{j}\in P$
*P*	Sequence of hovering points forming a tour, $P\subseteq P_{\mathrm{grid}}$
$P_{\mathrm{grid}}$	Set of all PDPs forming a grid
PDP	Possible Drone Points where the drone can hover to gather data
*r*	Radio range of the sensor nodes and the drone
$R_{i}$	The *i*-th tour created in the *i*-th iteration of the heuristics
$s_{i}$	A sensor node, $s_{i}\in S$
*S*	Set of all sensor nodes
$S_{\mathrm{subsets}}$	Set of subsets of sensor nodes; a set for each hovering point in *P*
$T_{\mathrm{allnodes}}$	Time to drones receive data from all sensor nodes, without considering the drone movement
$T_{\mathrm{collecting}}$	Collecting time: it is the time the drone spends hovering to gather data
$T_{\mathrm{moving}}$	Time the drone spends receiving data when it is moving
$T_{\mathrm{total}}$	Overall data gathering time
$T_{\mathrm{trip}}$	Trip time, is the time the drone spend in movement
TG	Trip Graph, *TG* = ($P_{\mathrm{grid}}$, *F*). *F* is the set of edges linking PDPs, the weights are the Euclidean distance
trs	The minimal distance the drone has to fly inside the area covered by the radio of a sensor node to enable this node to send all *m* bits of data to the drone or to relay nodes
*V*	Vertices representing all sensor nodes and the PDPs in the Connecting Graph
$w_{j}$	Weight attributed to PDP${}_{j}$ according to Equation (Equation 3)

**Table 2 sensors-20-06954-t002:** Characteristics of each scenario.

Scenario	Monitored Area (m${}^{2}$)	Number of Sensor Nodes	Drone Speed (m/s)	Data in Each Sensor Node (kbits)
1	200	30	2	20 to 120
2	400	150	2	20 to 120
3	400	100 to 250	2	60
4	400	150	0.5 to 3.0	60

**Table 3 sensors-20-06954-t003:** *t*-Test for Scenario 1.

Heuristics	Data Stored in Each Sensor Node					
	20	40	60	80	100	120
**Incremental**	29.82	30.15	30.29	19.83	18.37	17.63
**Decremental**	21.29	24.07	23.99	24.80	25.75	25.88

**Table 4 sensors-20-06954-t004:** *t*-Test for Scenario 2.

Heuristics	Data Stored in Each Sensor Node					
	20	40	60	80	100	120
**Incremental**	31.64	28.74	32.52	34.87	40.80	44.53
**Decremental**	23.49	34.69	40.04	34.90	34.86	34.08

**Table 5 sensors-20-06954-t005:** *t*-Test for Scenario 3.

Heuristics	Number of Sensor Nodes			
	100	150	200	250
**Incremental**	26.35	32.52	34.95	50.24
**Decremental**	24.22	40.04	44.18	45.87

**Table 6 sensors-20-06954-t006:** *t*-Test for Scenario 4.

Heuristics	Drone Speed					
	20	40	60	80	100	120
**Incremental**	25.60	27.59	31.02	32.52	32.99	35.97
**Decremental**	25.36	30.34	36.50	40.04	35.96	34.06

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
