# Peer review of "Gathering Big Data in Wireless Sensor Networks by Drone†"

_sensors, 2020, doi:10.3390/s20236954_

Round 1

Reviewer 1 Report

This manuscript documents an improved study on the use of mobile sinks (drone) for data collection in wireless sensor networks, by considering data gathering during both the drone hovering and movement to reduce the overall drone data gathering time. This is a fine contribution to the literature on this subject and is definitely worthy of publication. Overall, the manuscript is well written. I have some corrections/comments that I think will clarify the text:

  1. Line 467-469. It is described that the time the drone spends receiving data during its movement (T_moving) reduces the time the drone spends hovering. The reviewer is wondering how T_moving is varying and contributing to the reduction of the overall drone data gathering time? This would help to highlight the advantage of this newly proposed method.
  2. Line 481-484. In this work, the authors employed the Concorde Solver to solve the TSP, rather than using the original heuristics with a brute force algorithm that increased the execution time for larger monitored area too much. The reviewer is wondering how much time is saved by using this new Solver for different scenarios. This is important and needs clear clarification, as the conclusion ‘Our simulated experiments showed the proposed algorithm reduces up to 30% of the overall data gathering’ might not be fully valid, as part of contribution may come from the improvement of the use of the solver, rather than the newly proposed method.

Reviewer 2 Report

The paper presents and analyzes through simulation different algorithms for finding the best drone tour plan for gathering data from nodes in a WSN.

The paper is well written and has merit, although I found it hard to follow in some places due to the large number of notations used. 

I have only a few comments regarding the technical part. Maybe it would be better to explain in the introduction what "large volume of data" means in the context of the current paper? Defining the maximum data quantity from the start of the paper would present the problem in a clearer way. Another point would be to validate the simulation results in a real scenario. Detailing what would this require could be of interest for the readers.

Figures are oversized and, in my opinion, they could be minimized without loss of clarity and information.

Minor typos: "has" instead of "have" on line 65 on page 2 and "an edge" instead of "a edge" on line 310 on page 13.
